



# Whole soil warming decreases abundance and modifies community structure of microorganisms in subsoil but not in surface soil

Cyrill U. Zosso[1], Nicholas O.E. Ofiti[1], Jennifer L. Soong[2], Emily F. Solly[3], Margaret S. Torn[2], Arnaud Huguet[4], Guido L.B. Wiesenberg[1] and Michael W.I. Schmidt[1]

[1]Department of Geography, University of Zurich, Zurich, Switzerland
[2]Climate and Ecosystem Science Division, Lawrence Berkeley National Laboratory, Berkeley, CA, USA
[3]Group for Sustainable Agroecosystems, Department of Environmental Systems Science, ETH Zurich, Zurich, Switzerland
[4]Sorbonne Université, CNRS, EPHE, PSL, UMR METIS, Paris, France

*Correspondence to*: Cyrill U. Zosso (cyrill.zosso@geo.uzh.ch)

**Abstract.** The microbial community composition in subsoils remains understudied and it is largely unknown whether subsoil microorganisms show a similar response to global warming as do microorganisms at the soil surface. Since microorganisms are key drivers of soil organic carbon decomposition, this knowledge gap causes uncertainty in predictions of future carbon cycling in the subsoil carbon pool (>50 % of the soil organic carbon stocks are below 30 cm soil depth).

In the Blodgett forest field warming experiment (California, USA) we investigated how +4°C warming the whole soil profile to 100 cm soil depth for 4.5 years has affected the abundance and community structure of microorganisms. We used proxies for bulk microbial biomass carbon (MBC) and functional microbial groups based on lipid biomarkers, such as phospholipid fatty acids (PLFAs) and branched glycerol dialkyl glycerol tetraethers (brGDGTs). Microbial biomass decreased and community composition changed with depth. Our results show that the concentration of PLFAs decreased with warming in the subsoil (below 30 cm) by 28 % but was not affected in the topsoil. Phospholipid fatty acid concentrations changed in concert with soil organic carbon. The microbial community response to warming was depth dependent. The relative abundance of actinobacteria increased in subsoil, and gram+ bacteria in subsoils adapted their cell-membrane structure to warming induced stress as indicated by the ratio of *anteiso* to *iso* PLFAs. Our results show for the first time that subsoil microorganisms can be more affected by warming as compared to topsoil microorganisms. These microbial responses could be explained by the observed decrease in subsoil organic carbon concentration in the warmed plots. A decrease in microbial abundance in warmed subsoils might reduce the magnitude of the respiration response over time. The shift in the subsoil microbial community towards more actinobacteria might disproportionately enhance degradation of previously stable subsoil carbon, as this group is able to metabolize complex carbon sources.

## 1 Introduction

Soil temperatures are predicted to increase at all soil depths in near synchrony with air temperatures under global climate change (Soong et al., 2020a). Concentration driven projections of global air temperatures indicate a likely increase of 2.6-

4.8°C by 2100 under representative concentration pathway 8.5 (IPCC, 2013). It remains uncertain how this temperature increase will affect the soil organic carbon (SOC) pool. Soil organic carbon contains two to three times the amount of carbon present in the atmosphere (SOC ~ 1500-2400 PgC) (IPCC, 2013) and subsoils (below 30 cm soil depth) store around 50 % of the total SOC (Jobbagy and Jackson, 2000). Microorganisms are key players in the SOC cycle, but in subsoils there still exist
large uncertainties on how the microbial community will respond to warming (Jansson and Hofmockel, 2020). Environmental constraints for microorganisms are different in subsoils and topsoils, including plant inputs, SOC availability and stabilization processes, soil moisture or spatial heterogeneity (Rumpel and Kögel-Knabner, 2011). These constraints in subsoils generally lead to less microbial biomass and a microbial community which is adapted to a lower carbon availability (Brewer et al., 2019; Fierer et al., 2003b). Thus, topsoil and subsoil microbial communities may respond differently to
warming, with unknown effects on subsoil SOC stocks.

So far only a few field studies have been conducted to test the temperature sensitivity of subsoil microbial activity *in situ* (Hicks Pries et al., 2017; Nottingham et al., 2020), whereas there have been many incubation and mesocosm studies (e.g. Bai et al., 2019; Fierer et al., 2003a; Lin et al., 2018). The latter indicate that subsoil respiration is equally or more sensitive to
warming compared to topsoils (Bai et al., 2019; Fierer et al., 2003a; Lin et al., 2018), consistent with *in situ* observations of field warming experiments (Hicks Pries et al., 2017; Nottingham et al., 2020). The underlying microbial processes in subsoils are largely unexplored. To the best of our knowledge only Zhang et al. (2015) investigated how *in situ* warming for several years affected subsoil microbial biomass and community structure in a grassland experiment. They reported that warming increased microbial biomass throughout the whole soil profile and observed a depth-related change in the microbial
community structure (Zhang et al., 2015). In topsoils, microbial abundance tends to respond to warming differently depending on the ecosystem (Chen et al., 2015). Consequently, data from non-grassland ecosystems is needed to see whether the responses reported by Zhang et al. (2015) can be confirmed at other sites.

Processes controlling the microbial response to warming are still subject to intense scientific debate (Walker et al., 2018). An
interplay of acclimation and substrate depletion has been proposed to control the microbial response to warming (Melillo et al., 2017; Pold et al., 2017; Walker et al., 2018). Acclimation to warming encompasses physiological adaptations of individual microorganisms and shifts in the microbial community composition (Bradford et al., 2008). Substrate depletion on the other hand is a consequence of increased microbial activity with warming, which causes carbon limitation and in turn negatively affects microbial processes (Kirschbaum, 2004). But how these processes control the warming response of
microorganisms in subsoils remains largely unexplored until now.

We can assess microbial biomass and functional community composition using several parameters. Microbial biomass carbon (MBC) quantified by chloroform fumigation extraction is a proxy for overall microbial biomass (Vance et al., 1987), but does not yield any information on the microbial community composition. Phospholipid fatty acids (PLFAs) are microbial



membrane lipids that can be used to quantify microbial biomass, but also to describe the microbial community composition at a low taxonomic resolution (Willers et al., 2015). Furthermore, ratios among specific PLFAs have been used as indicators of environmental stress in soil (Wixon and Balser, 2013). For example, the ratio of *anteiso* to *iso* branched PLFAs is used as an indicator of a physiological adaptation of the microbial membrane of gram positive (gram+) bacteria to stress, among others caused by changes in temperature (Wixon and Balser, 2013). Adjusting the abundance of these compounds in their membrane helps microorganisms to control the fluidity of the membrane to maintain their activity under changing environmental conditions (Hall et al., 2010). We chose PLFA analysis over techniques with a higher resolution because of the precision between replicates and the high statistical power in differentiating experimental treatment effects (Ramsey et al., 2006). Core lipids of branched glycerol dialkyl glycerol tetraethers (brGDGTs) are an independent and complementary molecular marker that can also be used to quantify microbial abundance. The brGDGTs turn over on the order of decades (Weijers et al., 2010), whereas PLFAs cycle much faster (Kindler et al., 2009). The extracted core brGDGTs represent 10-20 % living biomass and the rest comes from microbial necromass (Gocke et al., 2017). This combination of molecular proxies for microbial abundance has not been explored so far in the context of manipulative soil warming experiments.

In this study, we make use of a soil warming experiment that warms the soils by +4 °C above ambient temperature down to 1 m depth at the University of California Blodgett Experimental Forest (Sierra Nevada, CA). We quantified how a temperature increase throughout the soil profile influenced the microbial biomass and community structure, using microbial biomarkers.

We hypothesized that warming would lead to more microbial biomass in top- and subsoils. Warming increased soil respiration at all depths at the study site (Hicks Pries et al., 2017; Soong et al., in press), which might be partially reflected in more microbial biomass. This would be in line with findings that in topsoil microbial biomass generally increases over the first years of soil warming, despite microbial biomass in forest soils being less responsive compared to other ecosystems (Chen et al., 2015). As previously reported for soil profiles we expect that the microbial community composition of top- and subsoils is also different at our site, because the subsoil microbial community is adapted to carbon scarcity and other environmental constraints (Brewer et al., 2019; Fierer et al., 2003b). We hypothesized that warming-induced changes in the microbial community composition will be depth-dependent. A meta-analysis proposed that the abundance of actinobacteria increases with warming (Chen et al., 2015). Actinobacteria are relatively more abundant in subsoils compared to topsoils (Fierer et al., 2003b) and an increase of this group would affect the subsoil microbial community more as compared to the topsoil community.



# 2 Materials and methods

## 2.1 Experimental setup and sampling

A whole-soil warming experiment was initiated in 2013 at the University of California Blodgett Experimental Forest, Sierra Nevada, CA (120°39'40" W; 38°54'43" N) to study how soil warming affects biogeochemical cycles in a natural environment (Hicks Pries et al., 2017). The site is located in a Mediterranean climate with mean annual precipitation of 1660 mm and a mean annual air temperature of 12.5 °C. The soil was classified as Alfisol of granitic origin and has a developed O-horizon. The experiment is located in a coniferous forest with ponderosa pine (*Pinus ponderosa*), sugar pine (*Pinus lambertiana*), incense cedar (*Calodefrus decurrens*), white fir (*Abies concolor*) and douglas fir (*Pseudotsuga menziesii*) as dominant tree species. The experiment consists of three blocks, each containing a pair of circular control and warmed plots of 3 m diameter. Temperature was maintained at +4°C from 0.2-1 m and +2.6 °C above 0.2 m in warmed compared to control plots (Hicks Pries et al., 2017; Soong et al., in press). Diurnal and seasonal temperature variations were considered depth-specifically (Hicks Pries et al., 2017; Soong et al., in press). Soil samples were collected in April 2018 after 4.5 years of warming with a 4.78 cm diameter manual soil corer. Organic horizons were removed prior to sampling. Mineral soil samples were recovered sequentially in 10 cm increments down to 90 cm depth and stored at 4 °C for transportation to the laboratory. In the laboratory, samples were freeze-dried, sieved to 2 mm and fine roots were removed before conducting subsequent analyses. Carbon concentrations were measured on an elemental analyzer coupled to an isotope ratio mass spectrometer (EA-IRMS; Flash 2000-HT Plus, linked by Conflo IV to Delta V Plus IRMS, Thermo Fisher Scientific, Inc.) on a ground subsample. The pH was measured on a dried subsample in 0.01 M $CaCl_2$. A second core was sampled from each plot using the same methods and stored at 4 °C until chloroform fumigation extraction was conducted during the week after sampling.

## 2.2 Microbial parameters

### 2.2.1 Microbial biomass carbon

Microbial biomass carbon (MBC) was analyzed using chloroform fumigation extraction, which is a method that measures the carbon released upon lysis of microbial cells (Vance et al., 1987). The extraction was conducted within one week after sampling on field moist soil, stored at 4 °C until processing. The analysis was conducted on the 0-10, 10-20, 30-40, 50-60 and 80-90 cm depth increments, using 10 g of the field-moist soil. Microbial biomass carbon was calculated as the difference in total organic carbon between subsamples fumigated with chloroform for five days prior to extraction with 0.05 M $K_2SO_4$ and subsamples directly extracted with 0.05 M $K_2SO_4$. Dissolved organic carbon was measured on a Shimadzu TOC-L (Shimadzu Scientific Instruments, Inc.) and calculated on a soil dry mass basis. A correction factor of 0.45 was used to account for carbon losses inherent to the method and make data comparable to previously published results (Vance et al., 1987). The difference in MBC between warmed and control plots was calculated as the difference between the warmed and control treatments normalized to the control plot within each of the three replicate paired blocks.



### 2.2.2 Phospholipid fatty acids

Analysis of PLFAs was performed following the method by Frostegård et al. (1991) with some modifications (Zosso and Wiesenberg, 2021). Briefly, we increased the amount of freeze-dried soil sample with soil depth, to account for the decreasing microbial biomass (from 3 g in the top soil to 12 g in deepest depth intervals). Using a solution of 1:2:0.8 (v/v/v) of chloroform : methanol : citric acid buffer, the extraction was repeated four times and the extract combined in a separation funnel. After addition of citric acid buffer and chloroform, the phases separated overnight and the organic phase was released thereafter. We added chloroform three more times and combined the organic phases. The extract was then separated in neutral-, glyco- and phospholipid fractions using a column with activated silica gel and the phospholipid fraction was methylated using boron trifluoride-methanol solution (Wiesenberg and Gocke, 2017). Prior to methylation deuterated eicosanoic acid ($D_{39}C_{20}$, Cambridge Isotope Laboratories, Inc.) was added as an internal standard for quantification to nmol g$^{-1}$soil. Quantification was performed using a Gas Chromatograph (GC, 7890 B, Agilent Technologies, Inc.) equipped with a multimode inlet (MMI) and a flame ionization detector. External standard series of 24 fatty acid standards and sample measurements on a GC (6890 N, Agilent Technologies, Inc.) coupled to a mass spectrometer (5973 N, Agilent Technologies, Inc.) followed by comparison to the NIST and Wiley mass spectra library supported peak identification. Both GCs were equipped with J&W DB-5MS columns (Agilent Technologies, Inc.) with the following dimensions: 50 m length x 0.2 mm inner diameter x 0.32 μm film thickness. The GC oven and MMI temperature programs are given in the supplementary (Table A1).

For total abundance of PLFAs, saturated fatty acids ($C_{14:0}$, $C_{15:0}$, $C_{16:0}$, $C_{17:0}$, $C_{18:0}$) were included. They were not included in calculations of the relative abundances of microbial groups (mol%), because they are not specific for any microbial group and distorted some of the depth patterns. To assess the microbial community structure, the following grouping of PLFAs was used (Willers et al., 2015): gram positive bacteria (gram+; $iC_{14:0}$, $aC_{14:0}$, $iC_{15:0}$, $aC_{15:0}$, $iC_{16:0}$, $aC_{16:0}$, $iC_{17:0}$, $aC_{17:0}$), gram negative bacteria (gram−; $C_{16:1\omega5c}$, $C_{16:1\omega7c}$, $C_{16:1\omega9c}$, $C_{18:1\omega5c}$, $C_{18:1\omega11c}$, $cyC_{17:0}$, $cyC_{19:0}$), fungi ($C_{18:2\omega6c}$) and actinobacteria ($10MeC_{16:0}$, $10MeC_{18:0}$). Microorganisms can also adapt their membrane composition in response to various stress factors, confounding physiological adaptations and changes in community structure (Wixon and Balser, 2013). The ratio of *anteiso* to *iso* branched PLFAs (($aC_{14:0}$ + $aC_{15:0}$ + $aC_{16:0}$ + $aC_{17:0}$) / ($iC_{14:0}$ + $iC_{15:0}$ + $iC_{16:0}$ + $iC_{17:0}$)) can partly capture adaptations of the membrane composition within gram+ bacteria in response to stress, rather than changes in the community composition (Hall et al., 2010; Wixon and Balser, 2013). The difference in PLFAs between warmed and control plots was calculated as the difference between the warmed and control treatments normalized to the control plot within each of the three replicate paired blocks.

### 2.2.3 Branched glycerol dialkyl glycerol tetraethers

Samples were extracted using Soxhlet extraction with 93:7 (v/v) dichloromethane : methanol as described in Wiesenberg and Gocke (2017). The extract was then prepared for analysis of core lipid brGDGTs as described by Coffinet et al. (2014).



Briefly, the samples were separated over a column of activated aluminum oxide in two fractions. Branched glycerol dialkyl glycerol tetraethers were separated using high performance liquid chromatography equipped with an automatic injector,

coupled with mass spectrometry with an atmospheric pressure chemical ionization source (HPLC-APCI-MS, Shimadzu LCMS 2020, Shimadzu Corp.). Two Hypersil Gold silica columns in tandem (150 mm × 2.1 mm, 1.9 μm; Thermo Finnigan, Inc.), thermally-controlled at 40 °C were used for brGDGT analysis. Semi-quantification of brGDGTs was performed by comparing the integrated signal of the respective compound with the signal of a $C_{46}$ synthesized internal standard assuming their response factors to be identical (Huguet et al., 2013).

Roman numerals correspond to the different brGDGT structures presented in Fig. A1, where compounds denoted by an accent after the Roman numerals are 6-methyl isomers. The relative abundances of individual brGDGTs (weight%) were calculated by dividing individual compounds by the sum of all brGDGTs. The difference in brGDGTs between warmed and control plots was calculated as the difference between the warmed and control treatments normalized to the control plot within each of the three replicate paired blocks.

**2.3 Data analysis**

For all statistical analyses, R 3.6.3 (R Core Team, 2020) was used. We assessed normality, homoscedasticity and model fit using residual plots, Shapiro-Wilk test, Levene test and outlier tests. We log transformed data if we observed poor fit. Correlation and multiple regression analysis were conducted to test the relation between MBC, PLFA, brGDGT and carbon concentrations. To test how warming affected microbial abundance and community, mixed effect models were built using

the *gls* and *lme* functions in the *nlme* package. We used restricted maximum likelihood with the random effect block (n=3) and the fixed effects treatment, depth and their interaction. We used Akaike's information criterion (AIC) to test whether variance structures improved model fit. If we found significant treatment*depth interactions, we employed Post hoc analysis and the model was run on a subset of depth increments. This is indicated by the specification of depth intervals included in the analysis. We reported results as significant at a level of p<0.05. In all figures and tables, we show the middle point of the

analyzed depth interval.





# 3 Results

## 3.1 Bulk parameters

The pH decreased with depth from 5.2±0.2 at 10-20 cm depth to 4.5±0.2 at 80-90 cm depth (p<0.01, Table 1), and was not different between control and warmed plots (p=0.814). Soil organic carbon concentrations decreased with depth from 66.9 mg g$^{-1}$ soil in the top 10 cm to 2.6 mg g$^{-1}$ soil at 80-90 cm depth (p<0.01, Table 1) and were lower in warmed compared to control plots below 20 cm (p=0.014) in the soil cores analyzed here and as described in more detail on replicated cores in Soong et al. (in press). Fine root concentrations did not change with depth (p=0.055) but were lower in warmed compared to

control plots (p=0.017; Table 1; Ofiti et al. (resubmitted)).

Table 1: pH, concentrations of soil organic carbon (SOC, mg g$^{-1}$ soil; Ofiti et al., resubmitted) and fine root mass (g m$^{-2}$, Ofiti et al., resubmitted) in control and warmed plots (mean± SE, n=3). More details on carbon concentrations on replicated cores in Soong et al. (in press).

| Soil depth [cm] | pH in 0.01M CaCl$_2$ | | SOC [mg g$^{-1}$ soil] | | Fine root mass [g m$^{-2}$] | |
|---|---|---|---|---|---|---|
| | **Control** | **Warmed** | **Control** | **Warmed** | **Control** | **Warmed** |
| -5 | - | - | 71.3±10.9 | 60.4±6.7 | 111.5±32.2 | 92.9±18.6 |
| -15 | 5.3±0.3 | 5.1±0.2 | 32.5±8.6 | 53.2±18.6 | 232.3±56.5 | 185.8±37.2 |
| -25 | 5.2±0.3 | 4.9±0.2 | 23.3±7.8 | 14.5±3.2 | 167.3±0.0 | 204.4±37.2 |
| -35 | 4.9±0.4 | 4.9±0.3 | 13.2±3.9 | 7.1±1.3 | 223.0±0.0 | 130.1±37.2 |
| -45 | 4.7±0.4 | 4.7±0.1 | 7.5±1.5 | 6.1±0.4 | 185.8±49.2 | 130.1±49.2 |
| -55 | 4.6±0.4 | 4.6±0.1 | 4.7±1.1 | 3.7±1.0 | 92.9±18.6 | 92.9±18.6 |
| -65 | 4.6±0.4 | 4.0±0.3 | 5.0±0.2 | 2.5±0.4 | 130.1±74.3 | 92.9±18.6 |
| -75 | 4.4±0.5 | 4.6±0.1 | 6.8±3.5 | 2.3±0.2 | 167.3±32.2 | 89.2±22.3 |
| -85 | 4.5±0.4 | 4.5±0.1 | 3.1±0.8 | 2.2±0.1 | 111.5±0.0 | 65.0±24.6 |

## 3.2 Microbial biomass carbon

Microbial biomass carbon (MBC) concentrations decreased with increasing depth in our soil cores, independent of treatment (p<0.01, Table A2). In the top 10 cm concentrations were 65.1±7.2 mmol C g$^{-1}$ soil and decreased to 15.0±5.3 mmol C g$^{-1}$ soil at 50-60 cm. Microbial biomass carbon concentrations were below the detection limit for the depth increment 80-90 cm. Warming did not affect MBC (p=0.564, Fig. 1). Concentrations of MBC and carbon concentrations were well correlated (R$^2$=0.62, Fig. A2).



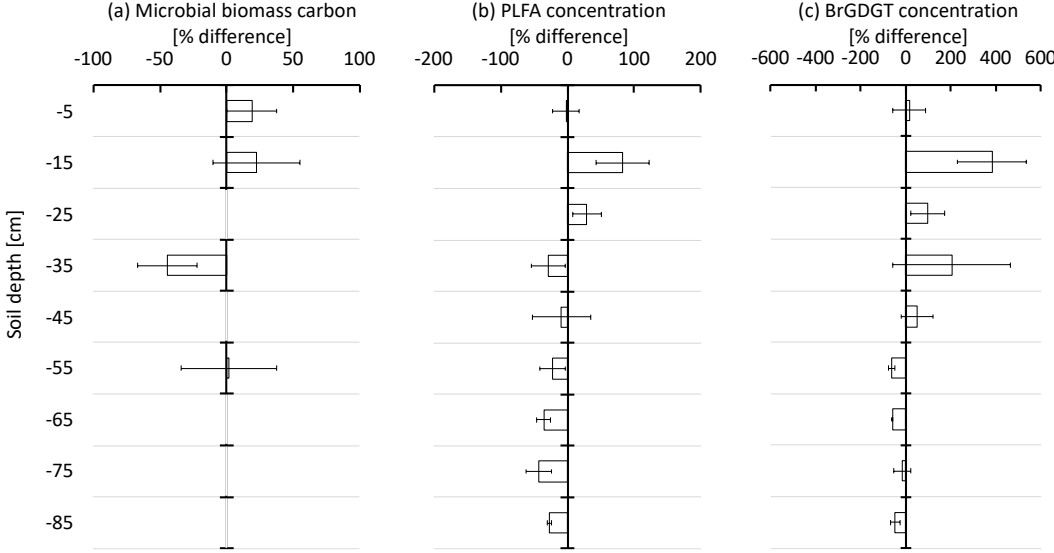

**Figure 1: Changes in microbial abundance between warmed and control plots based on (a) microbial biomass carbon using chloroform fumigation extraction (Soong et al., in press); (b) PLFA concentration and (c) brGDGT concentration after 4.5 years of +4°C soil warming. Values are shown as [(warmed-control)/control*100] (mean ± SE, n=3). The scale is adjusted to the respective values.**

## 3.3 Phospholipid fatty acids

Similar to the MBC, PLFA concentrations decreased with depth, independent of treatment (p<0.01, Table A2). Concentrations were 58.5±8.1 nmol g$^{-1}$ soil in the top 10 cm of the mineral soil and decreased to 2.7±0.2 nmol g$^{-1}$ soil at 80-90 cm. Concentrations of PLFAs and carbon as well as concentrations of PLFA and MBC were well correlated (R$^2$=0.88 and R$^2$=0.68, respectively; Fig. A3a,b). The proportion of PLFAs to SOC was higher with increasing soil depth (p<0.01, Fig. 2).





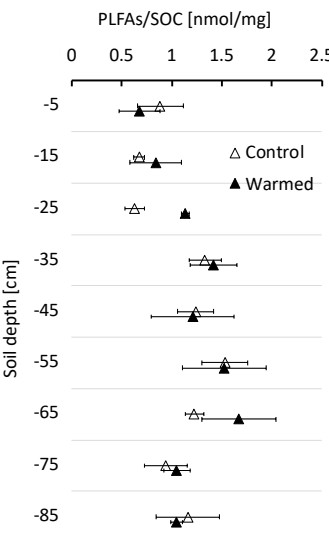


**Figure 2: Proportion of PLFAs to SOC in control and warmed plots after 4.5 years of +4°C soil warming (mean ± SE, n=3).**

The relative abundance of gram− bacteria decreased from 61.8±1.8 mol% in the top 10 cm of the mineral soil to 49.1±0.5
mol% at 80-90 cm and fungi from 7.4±1.2 mol% to 2.9±0.5 mol% ($p<0.01$ for both, Table A3). Gram+ bacteria and

actinobacteria on the other hand increased in relative abundance with depth from 24.6±1.2 mol% in the top 10 cm of the
mineral soil to 34.4±0.5 mol% at 80-90 cm and from 6.3±0.4 mol% in the top 10 cm of the mineral soil to 13.7±0.5 mol% at
80-90 cm, respectively ($p<0.01$ for both, Table A3). The ratio of *anteiso* to *iso* PLFAs decreased with depth ($p<0.01$, Fig. 3).



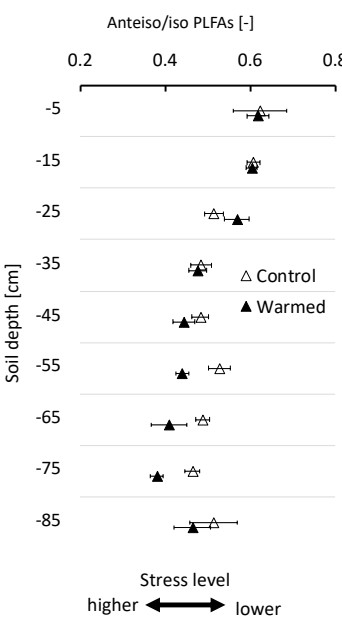

**Figure 3: The ratio of *anteiso* to *iso* PLFAs in control and warmed plots after 4.5 years of +4°C soil warming (mean ± SE, n=3). This ratio has been interpreted as an indicator of stress on gram+ bacteria with lower values indicating a higher stress level.**

The concentration of PLFAs was not affected by the warming treatment over the whole soil profile (p=0.057, Fig. 1b), however there was a significant interaction between treatment and depth (p=0.020). In the top 10 cm of the mineral soil there was no effect of warming on the concentration of PLFAs (p=0.843). Between 10 and 30 cm, there was a trend to higher concentrations of PLFAs in warmed compared to control plots (10-20 cm p=0.082 and 20-30 cm p=0.460). Running the mixed model with all depth increments below 30 cm, the concentration of PLFAs below 30 cm was significantly lower by 28% in warmed compared to control plots (p=0.002). The proportion of total PLFA concentration to SOC was not affected by warming (p=0.544, Fig. 2).

There were no differences between warmed and control plots in the relative abundance of gram+, gram−, actinobacteria and fungi (p=0.323, p=0.766, p=0.635 and p=0.179, respectively, Fig. 4). We further investigated the depth-specific effect of warming for actinobacteria due to the low significance level of the interaction of treatment and depth (p=0.056). Below 50 cm the relative abundance of actinobacteria was higher in warmed compared to control plots (p=0.004, Fig. 4a). The ratio of *anteiso* to *iso* PLFAs was lower in warmed compared to control plots (p=0.022, Fig. 3).



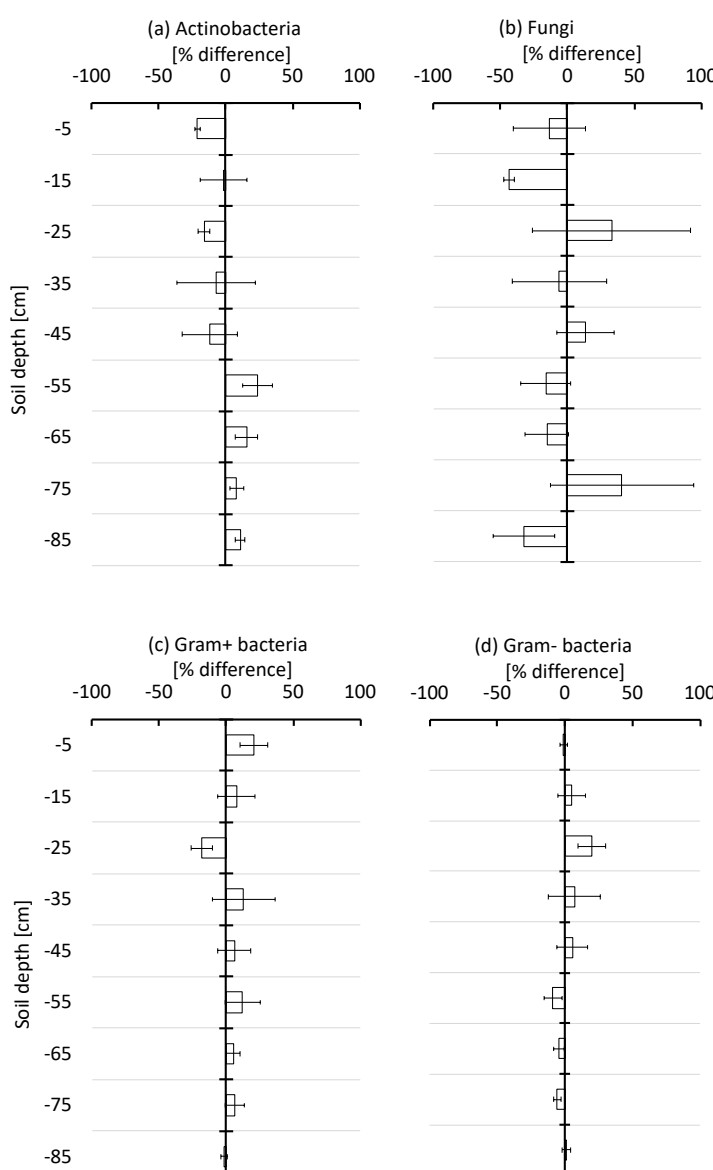

**Figure 4: Changes in the relative abundance of (a) actinobacteria; (b) fungi; (c) gram+ bacteria and (d) gram- bacteria based on PLFA biomarkers after 4.5 years of +4°C soil warming. Values are shown as [(warmed-control)/control*100] (mean ± SE, n=3).**

### 3.4 Branched glycerol dialkyl glycerol tetraethers

Concentrations of brGDGTs decreased with depth (p<0.01; Table A2). In the top 10 cm, concentrations were 68.7±23.7 ng g$^{-1}$ soil and decreased to 13.8±4.6 ng g$^{-1}$ soil at 80-90 cm depth. There was only a moderate correlation between brGDGT concentrations and concentrations of SOC, PLFA, or MBC ($R^2$=0.45, $R^2$=0.42 and $R^2$=0.33, respectively; Fig. A4a-c).

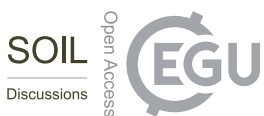

The relative abundance of individual brGDGTs compared to the total was dominated by compounds Ia (30.8±0.8 %), IIa (19.0±1.1 %), IIa' (21.2±1.1%), IIIa (9.5±0.5 %) and IIIa' (7.1±1 %, Table A4). IIa and IIIa were more abundant with depth

compared to the other compounds (p=0.002 and p=0.038, respectively), whereas the opposite was true for IIa' and IIIa' (p<0.01 and p<0.01, respectively; Table A4).

There was no effect of warming on the concentration of brGDGTs over the whole soil profile (p=0.422, Fig. 1c). Concentrations of brGDGTs were higher in warmed plots at 10-20 cm (p=0.041), but not affected by warming at 0-10 cm

(p=0.649) and the 10 cm increments from 20 to 50 cm (p=0.531, p=0.957 and p=0.801, respectively). The significance level of the interaction between treatment and depth was rather low (p=0.093) and upon visual inspection of the graph we ran the linear mixed model on all increments below 50 cm. Below 50 cm brGDGT concentrations were lower in warmed compared to control plots (p=0.027). There was no effect of warming on the relative abundance of any individual brGDGT (p>0.05; Table A4).

**4 Discussion**

**4.1 Depth trend in microbial abundance and community structure**

Both MBC and PLFA microbial abundance proxies indicated lower microbial concentrations in subsoils compared to topsoils, independent of treatment (Table A2). This depth trend strongly correlates with SOC concentrations (Fig. A2 and A3a), supporting the consensus that the concentration of SOC is an important controlling factor on microbial abundance in

soils (Fierer, 2017; Soong et al., 2020b). Similarly, carbon availability is an important factor structuring the microbial community composition in soil depth profiles, with oligotrophic bacteria favoured when nutrients become limiting (Brewer et al., 2019; Fierer et al., 2003b). Increasing relative abundance of gram+ and decreasing abundance of gram− bacteria with depth (Table A3) indicates a decrease in SOC availability (Fanin et al., 2019). Gram− bacteria are more dependent on labile plant-derived carbon sources, whereas gram+ bacteria are able to grow on more complex carbon sources (Fanin et al., 2019).

The observed relative increase in actinobacteria with depth (Table A3) is consistent with previous findings from a grassland site (Fierer et al., 2003b). It has been shown that certain actinobacteria are able to degrade complex carbon sources and grow mycelia-like structures, which might allow this microbial group to thrive in the subsoil (Goodfellow and Williams, 1983). Fungal abundance decreased with depth (Table A3), likely because fresh plant-derived carbon, which supports the growth of saprotrophic fungi, is more available in the upper soil horizons (Griffiths et al., 1999; Lindahl et al., 2007). Finally, the ratio

of *anteiso* to *iso* fatty acids has been used as a stress indicator in gram+ bacteria (Wixon and Balser, 2013), where a lower ratio indicates that the membrane fluidity is higher, as an adaptation of microorganisms to maintain growth (Hall et al., 2010). The lower ratio in subsoils (Fig. 3) might be related to the low-carbon environment in subsoils. However, other factors such as pH, soil moisture, or temperature also shape the microbial community (Drenovsky et al., 2004; Fierer, 2017).



For example, the pH at our site is lower in the subsoil compared to topsoil, which most likely will stimulate adaptations of the cell membranes.

The concentrations of brGDGTs also decreased with depth (Table A2), but the correlation with carbon concentrations (Fig. A4a) and the other microbial biomass proxies (Fig. A4b,c) was only moderate ($R^2 < 0.45$). The moderate correlation is likely because brGDGTs are a time-integrated signal of accumulated necromass with a turnover time of several decades (Weijers et al., 2010). We did observe significant depth trends for several brGDGTs. However, it is unclear whether these trends reflect changes in the community structure or physiological adaptations of single microorganisms with depth as the source organisms are not yet fully identified (Sinninghe Damsté et al., 2014).

Overall, carbon concentrations seem to be a strong control on the microbial proxies MBC and PLFAs, whereas brGDGT distribution is known to be strongly influenced by mean annual air temperature and pH, but may also be affected by other factors like vegetation (Ayari et al., 2013), seasonality (Huguet et al., 2013) or soil moisture (Menges et al., 2014).

## 4.2 Less microbial biomass with warming in subsoil, but not topsoil

Our results show a distinctly different response of topsoil and subsoil microbial abundance to warming in PLFA concentrations (Fig. 1b). There was no significant effect of warming on the microbial abundance in the topsoil, which confirms previous observations from forest topsoils (Chen et al., 2015). Lower PLFA concentrations in warmed compared to control subsoils contradict our hypothesis that the increased respiration observed at all depths (Hicks Pries et al., 2017; Soong et al., in press) is at least partially reflected by an increase in microbial biomass. Lower concentrations of brGDGTs in warmed compared to control subsoils are in line with the observation in PLFAs, even though only at depths from 50 to 90 cm (Fig. 1c). Due to the higher persistence of brGDGTs in soils there might be a lag in response as compared to PLFAs (Weijers et al., 2010). Microbial biomass carbon on the other hand was not affected by warming at any depth (Fig. 1a). We hypothesize that the bulk measurement of MBC might be less powerful at capturing treatment effects compared to the targeted molecular analysis of PLFAs and brGDGTs.

Limitation of available carbon has been suggested as a potential driver of decreasing microbial abundance in long-term warming experiments (Kirschbaum, 2004; Walker et al., 2018) and has been reported in surface soils of field warming experiments (Melillo et al., 2017; Pold et al., 2017). Microbial abundance closely tracked carbon concentrations with depth and as they changed with warming. The ratio of total PLFAs to SOC did not change with warming (Fig. 2), despite 28 % lower PLFA concentrations in the warmed subsoil (Fig. 1b). Soil organic carbon inputs to subsoils in the form of dissolved organic carbon (DOC) cascading down from the topsoil are often strongly processed (Roth et al., 2019) and fine root litter inputs decline with depth (Leppälammi-Kujansuu et al., 2013). Declining SOC inputs with depth may limit microbially available carbon with warming in subsoils as compared to topsoils. We observed lower concentrations of particulate organic

matter (Soong et al., in press) and fine root mass in the warmed subsoils (Table 1; Ofiti et al., resubmitted). After 3.5 years of warming, unprotected SOC (i.e., not mineral associated) had been depleted due to warming in the subsoil (Soong et al., in press), indicating a loss of easily accessible organic matter for subsoil microorganisms. These results suggest that in this

forest, carbon limitation might be one cause for the lower abundance of microorganisms in subsoil after 4.5 years of +4 °C warming.

In addition to SOC availability, temperature as such (Bradford, 2013) and changes in soil moisture associated with warming (Manzoni et al., 2012) likely affect microbial abundance. For example, temperature controls the reaction rates of microbial

enzymes, which in turn can affect microbial abundance (Allison et al., 2010). Thus, the higher magnitude of warming below 20 cm in our experiment might be partially responsible for the observed difference in the microbial response between top- and subsoil. Furthermore, warming of the soil caused a significant decrease in the volumetric water content by 2-5 percentage points throughout the soil profile in the warmed plots in our experiment (Soong et al., in press). Drying is expected to coincide with rising temperatures in Mediterranean climate regions under future climate scenarios (Polade et al.,

2017). In mesic soils, drying generally has a negative effect on microbial abundance and activity (Manzoni et al., 2012).

To our knowledge, only Zhang et al. (2015) reported microbial responses to *in situ* subsoil warming. In their experiment, warming induced an increase in microbial biomass down to 50 cm and the authors proposed that this was due to increased SOC availability due to stimulated plant growth, a reduction of anaerobic conditions, and better thermal growth conditions

for microorganisms (Zhang et al., 2015). The uniform increase in microbial biomass and the proposed underlying processes could not be observed in our experiment perhaps because the warming treatment was different. Zhang et al. (2015) created a gradient of warming from +1.4 °C at 10 cm depth to only +0.1 °C at 50 cm depth, barely warming the subsoil. Warming throughout the profile may offer a treatment closer to predictions of whole-soil warming by climate change (Soong et al., 2020a). Furthermore, plant dynamics and environmental factors such as temperature and soil hydrology are very different

between their grassland site on the Tibetan plateau and our forest site in Mediterranean climate. Microorganisms at their site are much more limited by temperature (MAT -3.8 °C) and affected by anoxic conditions (Zhang et al., 2015). We propose that similarly to topsoils, in subsoils the ecosystem type and experimental setup strongly affect the observed response of microbial abundance to warming (Chen et al., 2015).

**4.3 Small effect on microbial community composition in subsoils**

Whereas 4.5 years of +4 °C warming decreased subsoil microbial abundance by 28 %, there were only minor effects on the microbial community structure as assessed by organic geochemical methods. Only actinobacteria increased in relative abundance with warming in the lower subsoil (below 50 cm, Fig. 4a). We also observed a decrease in the ratio of *anteiso* to *iso* fatty acids with warming (Fig. 3). Changes in this ratio have been attributed to physiological adaptations of gram+ bacteria to warming-induced stress rather than shifts in the community structure (Wixon and Balser, 2013).






So far, laboratory incubations tentatively identified actinobacteria as one of the microbial groups that can be stimulated by warming (Oliverio et al., 2017). Our field observation of higher relative abundance of actinobacteria in the lower subsoils confirms findings from these laboratory incubations, and shows that warming affected actinobacteria less negatively relative to the other microbial groups. Furthermore, our results are also consistent with observations by Frey et al. (2008). They

reported increased relative abundance of actinobacteria in a long-term warming experiment in topsoils after more than 12 years of warming, where lower SOC availability was a main driver of the microbial response (Frey et al., 2008). Characteristics of actinobacteria indicate their ability to degrade complex polymers in soils (Goodfellow and Williams, 1983) and long-term experimental warming caused an increase in carbohydrate degrading genes of actinobacteria (Pold et al., 2016). Finally, actinobacteria seem to be more resistant to drought than other microbial taxa (Naylor et al., 2017). It

remains uncertain whether this is due to their ability for sporulation or other mechanisms (Naylor et al., 2017), but the resistance to decreasing soil moisture could further contribute to the relative increase in abundance in our warmed subsoils. Thus, actinobacteria might be relatively more abundant due to a combination of resilience to warming and adaption to lower SOC concentrations and reduced soil moisture.

Although warming did not alter the relative abundance of gram+ bacteria, it did significantly change the fatty acid composition of microbial membranes, expressed as the ratio of *anteiso* to *iso* fatty acids in the subsoil (Fig. 3). The ratio of *anteiso* to *iso* fatty acids likely reflects an acclimation to changing temperatures in gram+ bacteria, which might give gram+ bacteria a competitive advantage because they can adapt to changing environmental conditions (Hall et al., 2010). Nonetheless, the synthesis of new lipids is energy intensive and might contribute to the lower efficiency at which carbon is

incorporated to microbial biomass (Hall et al., 2010).

Multiple years of soil warming caused a shift in the microbial community at Blodgett Forest towards higher relative abundance of organisms able to utilize complex organic compounds and physiologically adapt to higher temperatures. Thus, there could be continued enhancement of decomposition including previously stable SOC in warmed subsoils.


The shifts in the microbial community and physiology observed by PLFAs were not reflected in the relative abundance of individual brGDGTs (Table A4). Likewise, at a decadal geothermal temperature gradient the brGDGT fingerprint only changed at high warming magnitude where soil temperatures were $> +14°C$, when the microbial community was also significantly altered (De Jonge et al., 2019). In our experiment, lower concentrations of brGDGTs in the subsoil (Fig. 1c) did

not affect the relative abundance of individual brGDGTs. This indicates that after 4.5 years of warming there was no shift in the microbial community or physiology which produced detectable shifts in brGDGTs, nor preferential decomposition of certain core brGDGTs.




Further research is needed to assess how microbial abundance and community composition will evolve with long-term
subsoil warming. A long-term study at Harvard Forest indicated that microbial response to warming, at least in topsoils,
progressed in cycles of change in the microbial community and decomposition rates (Melillo et al., 2017). It is not known
whether similar impacts will occur in subsoils. Furthermore, seasonality can affect microbial biomass and community
structure (Contosta et al., 2015) and only having samples from one time point is a limitation in our study. The fact that
subsoil microorganisms are differently affected by warming compared to topsoils is valuable information needed to predict
the long-term response of ecosystems to climate change, even if these effects diminish over the long term or are seasonally
varying.

## 5 Conclusion

Microbial biomass concentrations in warmed subsoil (below 30 cm) were 28 % lower compared to the control after 4.5 years
of +4 °C warming, whereas they were not significantly affected in topsoils, consistent with changes in SOC stock. We
showed for the first time that subsoil and topsoil microbial communities respond differently to whole soil warming. In the
lower subsoil (below 50 cm) actinobacteria were relatively more abundant, whereas gram+ bacteria revealed physiological
adaptations of the cell membrane in response to warming-related stress. Changes in the amount and composition of subsoil
SOC along with the direct effects of warming are likely main drivers of changes in the microbial community. Whether less
biomass in subsoils will lead to reduced respiration in the future remains to be investigated. The observed shift in the subsoil
microbial community composition favoured actinobacteria which are able to metabolize more complex carbon sources,
suggesting that soil warming may lead to microbial responses that disproportionately enhance degradation of previously
stable subsoil carbon.



**6 Appendices**

Appendix A-Figures

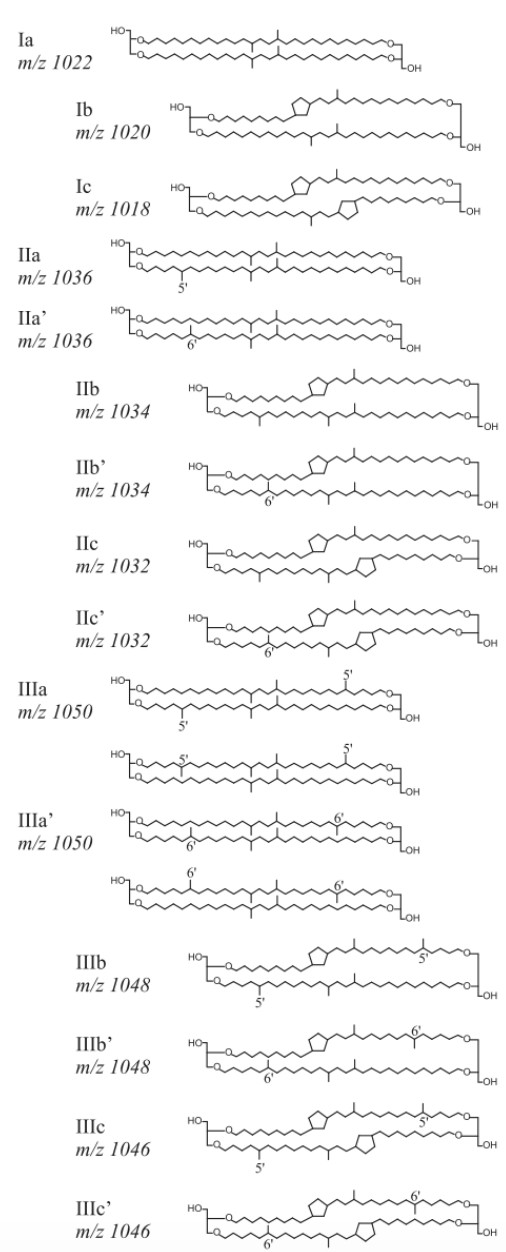

**Figure A1: Chemical structures and molecular masses of brGDGTs as described in De Jonge et al. (2014).**





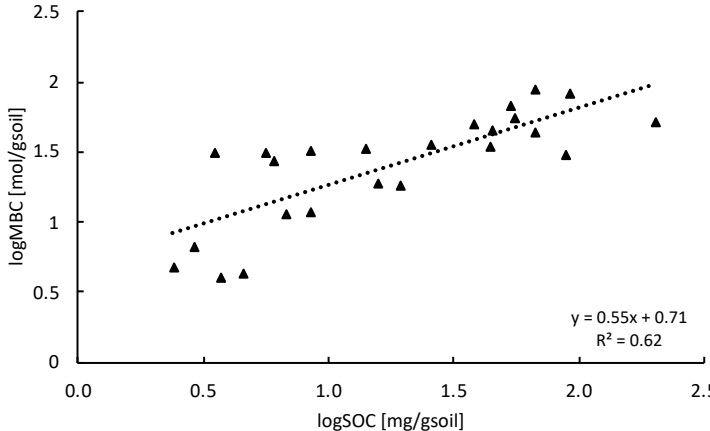


**Figure A2: Relationship between MBC and SOC at all depth increments (n=24). Values were log-transformed to meet normality.**





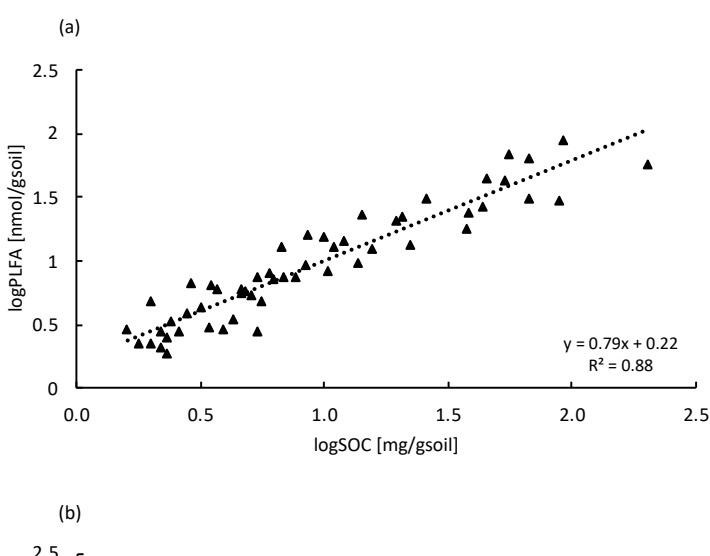

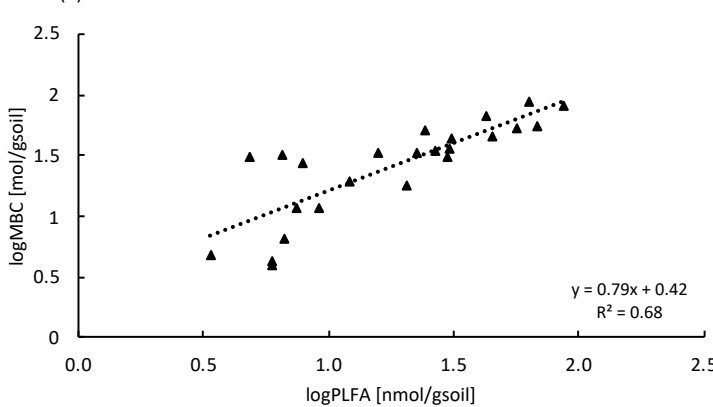


**Figure A3: Relationship between (a) PLFA with SOC (n=54) and (b) MBC with PLFA (n=24) at all depth increments. Values were log transformed to meet normality.**





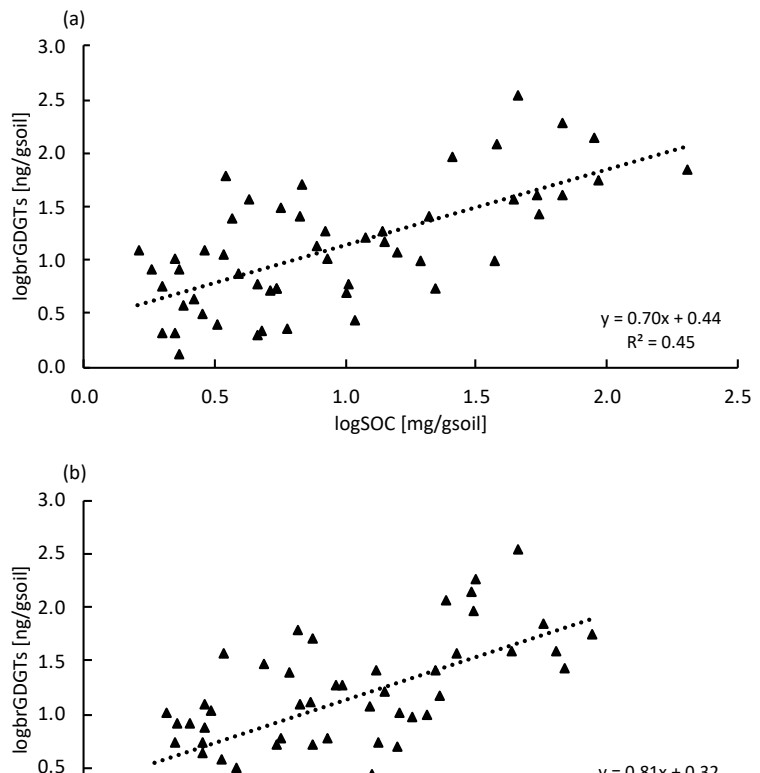


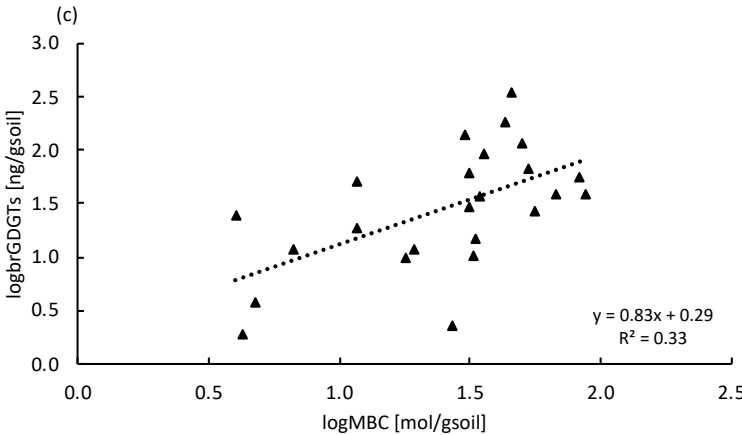

**Figure A4: Relationship between (a) brGDGT with SOC (n=54) (b) brGDGT with PLFA (n=54) and (c) brGDGT with MBC (n=24) at all depth increments. Values were log transformed to meet normality.**




Appendix B-Tables

**Table A1: GC oven and multimode inlet temperature programs for PLFA measurements.**

| GC oven temperature program | | | Multimode inlet temperature program | | |
|---|---|---|---|---|---|
| **Rate** | **Temperature** | **Hold time** | **Rate** | **Temperature** | **Hold time** |
| [°C min⁻¹] | [°C] | [min] | [°C min⁻¹] | [°C] | [min] |
| - | 50 | 4 | - | 60 | 0.5 |
| 10 | 150 | 0 | 850 | 400 | 5 |
| 2 | 160 | 0 | 50 | 250 | - |
| 0.5 | 170 | 10 | | | |
| 0.2 | 175 | 10 | | | |
| 0.2 | 180 | 10 | | | |
| 0.2 | 185 | 5 | | | |
| 0.2 | 190 | 5 | | | |
| 2 | 210 | 0 | | | |
| 5 | 320 | 15 | | | |




**Table A2: Concentrations of MBC using chloroform fumigation extraction (Soong et al., in press), PLFAs and brGDGTs in control and warmed plots (mean± SE, n=3).**

| Soil depth [cm] | MBC [mmolC g⁻¹ soil] | | PLFA [nmol g⁻¹ soil] | | brGDGT [ng g⁻¹ soil] | |
|---|---|---|---|---|---|---|
| | Control | Warmed | Control | Warmed | Control | Warmed |
| -5 | 60.6±11.5 | 69.5±10.3 | 62.4±16.6 | 54.5±6.2 | 88.7±48.1 | 48.8±9.8 |
| -15 | 34.8±9.0 | 37.5±4.5 | 21.1±4.5 | 35.2±5.0 | 55.2±31.8 | 190.5±77.3 |
| -25 | - | - | 13.2±2.7 | 16.3±2.9 | 6.9±1.3 | 14.7±6.5 |
| -35 | 26.1±4.4 | 16.3±8.5 | 17.1±4.6 | 10.3±2.9 | 8.9±3.6 | 10.2±4.8 |
| -45 | - | - | 9.6±3.1 | 7.7±3.0 | 6.7±3.2 | 10.5±7.6 |
| -55 | 15.7±8.2 | 14.2±8.5 | 6.7±0.4 | 5.0±1.0 | 45.0±10.7 | 15.2±7.7 |
| -65 | - | - | 6.2±0.6 | 4.0±0.6 | 5.4±0.3 | 2.2±0.1 |
| -75 | - | - | 5.5±2.2 | 2.3±0.3 | 9.7±4.6 | 4.6±2.0 |
| -85 | - | - | 3.1±0.2 | 2.3±0.1 | 19.7±8.2 | 7.9±1.4 |




**Table A3: Relative abundance of microbial groups, identified using PLFAs in control and warmed plots (mean± SE, n=3).**

| Soil depth [cm] | Gram+ [mol%] | | Gram- [mol%] | | Fungi [mol%] | | Actinobacteria [mol%] | |
|---|---|---|---|---|---|---|---|---|
| | Control | Warmed | Control | Warmed | Control | Warmed | Control | Warmed |
| -5 | 22.5±1.6 | 26.7±0.5 | 62.1±0.4 | 61.5±1.7 | 8.5±2.0 | 6.3±1.1 | 7.0±0.5 | 5.5±0.4 |
| -15 | 26.6±2.0 | 28.2±2.1 | 56.7±3.7 | 58.8±2.2 | 6.1±0.6 | 3.4±0.2 | 10.5±2.2 | 9.6±0.1 |
| -25 | 35.0±4.9 | 27.9±1.7 | 47.3±5.2 | 55.8±2.3 | 3.3±1.3 | 4.4±1.9 | 14.3±1.6 | 11.9±0.8 |
| -35 | 27.6±4.7 | 29.2±3.2 | 53.6±4.5 | 55.8±5.8 | 5.6±2.1 | 3.8±0.6 | 13.3±1.9 | 11.3±2.0 |
| -45 | 29.0±4.4 | 29.8±1.9 | 54.3±5.7 | 56.0±1.2 | 3.3±1.4 | 3.4±1.0 | 13.4±2.6 | 10.8±0.5 |
| -55 | 29.6±1.4 | 32.8±2.2 | 55.2±1.6 | 50.1±2.5 | 4.0±0.3 | 3.3±0.6 | 11.1±0.5 | 13.7±0.6 |
| -65 | 30.5±2.1 | 32.1±2.0 | 54.6±1.3 | 52.3±2.4 | 4.0±0.8 | 3.1±0.2 | 10.9±0.4 | 12.6±0.6 |
| -75 | 33.0±3.8 | 34.6±1.7 | 51.5±4.3 | 48.3±2.7 | 2.4±0.7 | 2.9±0.9 | 13.2±1.0 | 14.2±1.1 |
| -85 | 34.7±0.7 | 34.1±1.0 | 48.9±1.1 | 49.3±0.7 | 3.5±0.6 | 2.2±0.5 | 13.0±0.6 | 14.3±0.4 |




**Table A4: Relative abundance of individual brGDGTs compared to the total in control and warmed plots (mean± SE, n=3). All compounds were included for the calculation, but only the five most abundant are shown in the table.**

| Soil depth | Ia [%] | | IIa [%] | | IIa' [%] | | IIIa [%] | | IIIa' [%] | |
|---|---|---|---|---|---|---|---|---|---|---|
| [cm] | Control | Warmed | Control | Warmed | Control | Warmed | Control | Warmed | Control | Warmed |
| -5 | 24.8±7.7 | 29.4±2.6 | 14.8±4.6 | 16.8±3.2 | 26.3±4.4 | 25.3±2.0 | 11.2±0.9 | 9.9±4.0 | 14.6±7.5 | 8.6±3.1 |
| -15 | 25.3±4.7 | 30.4±2.5 | 15.9±6.8 | 20.1±5.0 | 26.6±4.0 | 25.2±3.7 | 9.4±3.7 | 8.8±2.4 | 16.2±10.0 | 8.1±2.8 |
| -25 | 28.6±3.0 | 35.8±1.6 | 11.9±4.7 | 17.0±3.7 | 28.3±2.8 | 21.9±7.9 | 6.7±1.6 | 7.2±1.8 | 9.4±2.4 | 5.2±2.2 |
| -35 | 32.1±2.7 | 33.9±5.4 | 11.9±2.9 | 17.1±3.0 | 28.1±1.7 | 19.1±9.5 | 7.0±1.0 | 6.8±0.8 | 8.4±1.4 | 3.8±1.8 |
| -45 | 30.7±2.7 | 35.1±3.4 | 15.1±3.9 | 23.9±3.9 | 25.7±4.5 | 16.5±3.6 | 8.4±0.1 | 9.8±2.3 | 7.3±2.1 | 3.4±0.7 |
| -55 | 31.0±0.5 | 33.0±4.7 | 21.3±8.2 | 27.4±6.4 | 21.7±6.4 | 15.2±5.7 | 10.6±2.7 | 12.2±1.7 | 5.4±1.9 | 3.3±0.6 |
| -65 | 33.4±2.0 | 31.4±9.4 | 20.2±7.0 | 19.8±3.8 | 19.3±6.2 | 15.2±4.0 | 10.2±3.2 | 10.5±2.3 | 5.1±1.3 | 3.7±0.6 |
| -75 | 32.5±1.9 | 31.6±10.6 | 22.5±7.7 | 20.8±2.4 | 18.2±7.6 | 15.2±6.0 | 9.8±3.2 | 11.3±2.2 | 4.7±1.9 | 3.1±0.8 |
| -85 | 28.1±1.0 | 31.3±7.3 | 24.8±8.5 | 26.6±3.0 | 17.0±8.4 | 12.0±3.6 | 12.2±3.9 | 11.9±1.8 | 4.7±1.9 | 2.4±0.5 |




## 7 Data availability

The data used in this study will be deposited on the ESS-DIVE repository (https://ess-dive.lbl.gov/) upon acceptance.

## 8 Author contributions

MST designed and maintained the warming field experiment. MWIS conceived the DEEP C project. CUZ, NOEO, JLS, EFS, GLBW, MST, MWIS participated in the field campaign. CUZ carried out biogeochemical analyses, supervised by GLBW and AH. All co-authors contributed to data interpretation and to the manuscript written by CUZ.

## 9 Competing interests

The authors declare that they have no conflict of interest.

## 10 Acknowledgments

The research was supported by Swiss National Science Foundation (SNF) project 172744 (DEEP C), U.S. Department of Energy Office of Science, Office of Biological and Environmental Research Terrestrial Ecosystem Science Program, under Award DE-SC-0001234 and the SATW Germaine de Staël project 2019-08. We thank C. Castanha (LBL) for field and T.
Kraut (UZH) for laboratory assistance. S. Derenne (Sorbonne Université) and M. van de Broek (UZH) contributed to the advancement of our work with valuable comments and discussions. The University Research Priority Program Global Change and Biodiversity (URPP-GCB) at the University of Zurich offered opportunities for helpful discussions and exchange.

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
