# Peer review of "Whole soil warming decreases abundance and modifies community structure of microorganisms in subsoil but not in surface soil"

_SOIL, 2021_

## Author Response (AR1)

Dear Dr. Ashish Malik

Following your response, we implemented the changes as outlined in our response. In our point-by-point reply below, you find in **bold** the editors/reviewers comments, in blue the initial response and in *italic green* a more detailed description of the changes implemented. Line numbers refer to the marked-up version of the manuscript. We hope you find the changes satisfying.

We thank you, as well as Grace Pold and the anonymous reviewer for your time and thoughts on our manuscript.

Yours Sincerely,

Cyrill Zosso

**Response to the Editors comment**
**I think the figures can be improved a bit. Results of statitical analysis can be presented to make it easy for the reader to judge those that are significant. In Figure 2 and 3, the yaxis is missing.**
We thank Dr. Ashish Malik for handling our manuscript and the comments, which have also been raised by the reviewers.

We will improve the figures, adding y axis as suggested. We will also try to make figure 4 more accessible as suggested by Grace Pold. Here, we already tried an ordination, however this did not improve it. We consider highlighting the most important parts of the figure to better guide the reader.

*We worked on the formatting of all figures and hope they are more visually appealing now. After testing various visualizations for figure 4, we decided to keep the same type of figure, although we reformatted it and hope it is easier to read this way.*

Concerning the statistical analysis, we will include the comments by Grace Pold, better describing and justifying the approach used. As we generally focused on the results from the LME in the discussion and did not further discuss the Post-hoc tests, we suggest to delete the two sections where we describe these tests (lines 226-228 and lines 254-255) to make the statistics more understandable.

*We highlight below how we made the statistical analysis more understandable, so it is easy to follow which results are significant.*

**Response to the interactive comment by Grace Pold**
Cyrill Zosso and colleagues present an interesting look into how deep soil warming alters microbial biomass and high-level community abundance in a forest ecosystem. Overall, I think this is a very interesting paper and applaud the authors for their use of lipids rather than getting stuck in the mud with sequencing, as it provides a different perspective on warming impacts on microbial communities than is typically used. The paper is also well-written and flows well, and was enjoyable to read. There are a few areas I think the manuscript could benefit from clarification on. In particular, how and/or why certain comparisons for depth*warming interactions were made, and a better integration and justification for the brGDTs in hypothesis testing.

We thank Grace Pold for the constructive and insightful comments. Below we provide our point-by-point reply and we believe the comments will help us further improve the manuscript.

**Main comments:**
**The scale on which the data are presented does not seem to match the scale on which the analyses were completed.**

We were not quite sure how to understand this comment. However, cross-checking all figure units and calculations again, we did find that there was a mistake in the labelling of MBC graph and table. The values are reported in mmolC/gsoil, but should be nmolC/gsoil. For comparability, we double-checked which is the most widely used unit. As it is more often the case that MBC data are presented in ugC/gsoil, we would suggest to report them in these units. It is then easier to compare our values with other publications. Nothing changes in the trends reported.

*We adapted all numbers concerning MBC to ugC/gsoil. More specifically, this concerns the results section on microbial biomass carbon (Chapter 3.2, lines 242ff), as well as in the supplement Figures A2, A3 (b), A4 (c) and Table A2.*

**It would be very helpful to post the R scripts with the manuscript. I found it hard to follow what was being included as a random effect (or whether random effects were nested) in the different results. Was a model of form "lipid ~warming *depth + (1|block/warm)" or "lipid ~ warming *depth + (1|block/depth)" fit? Or something else?**
Below we post the model used and we will also publish the script alongside the manuscript (as well as the dataset) on ESS-DIVE repository. We did consider taking plots (n=6) nested within block, however we did not have replication within blocks. We did not include warm or depth nested within block as we considered them fixed effects.

"lipid ~ warming*depth, random=~1|block"

*As mentioned we used the above model and will publish the script and dataset alongside the manuscript on ESS-DIVE repository upon acceptance.*

**Related, there seem to be a lot of post-hoc tests, but it is not always clear how these posthoc tests were selected for completion. Why is the top 10cm sometimes compared to the bottom 10cm, and other times some intermediate depth compared to the deep?**
We were not quite sure where the reviewer saw a comparison of top 10cm to bottom 10cm, but we think it could be the following misunderstanding. We think this could be related to e.g. lines 213-214. Here, we state the concentration in the top and in the lowest depth increment and a p-value. However, in this section we do not compare solely the top 10cm to bottom 10cm, but just mention these values as starting and end point. The p-value is from the LME model which was run through all depth points. If this was the problem, we suggest to reformulate these sections.

*We changed the formulation and split the information in two sentences, as to avoid any confusion (e.g. see lins 366).*

We only report post-hoc analysis including all depths for total PLFA concentrations (lines 226-228) and brGDGTs (lines 254-255), where we ran the function emmeans(). However, to simplify and reduce confusion, we propose to delete these two sections. These post-hoc tests are not discussed further in the discussion, since we generally focused on the results of the LME, which we explain in more detail below.

*We deleted these sections, but added more specification to which depths were included in the LMEs we ran (see below).*

**Or why is the cutoff for depth 30cm in some instances, and 50cm for others (ex. total PLFA vs. Actinobacteria)? Why is there only one p-value reported (ex. L229-230) when looking at all the depths below a certain point, and not one for each of the depths analyzed as show in the figure? Was the total lipid below a certain depth summed for each core to complete this analysis? [I think this would make sense from a statistics standpoint]**

We generally report p-values from LMEs (except Post-hoc mentioned above). The models were first run for the whole profile. We then additionally ran the LME model on a subset of depths (either below 30cm for total PLFA or below 50cm for actinobacteria/brGDGTs) if the treatment*depth interaction was significant or upon visual inspection of the plots. Furthermore, we observed more pronounced treatment effects in the lower subsoils for several other variables as reported in other papers, such as the average chain length of solvent extractable lipids ('*Ofiti et al. 2021, SBB, Warming promotes loss of subsoil carbon through accelerated degradation of plant-derived organic matter*') or d13C and free particulate organic matter ('*Soong et al. 2021, accepted for Science Advances, Five years of whole-soil warming led to loss of subsoil carbon stocks and increased CO2 efflux*'). That's why we think it is interesting that such effects were also observed in microbial parameters in the lower subsoil. Thus, the values reported in the manuscript are from the LME. When we sum up the values below a certain depth as suggested by the reviewer and run a t-test, the values are $p<0.01$, $t=-4.87$, $df=4$ for Actinobacteria and $p=0.054$, $t=2.7$, $df=4$ for brGDGTs (see table below). As this does not account for the blocks and depth, we suggest to continue reporting the values from the LME.

*We added one sentence in the methods section (line 222) on why we ran the LME on a subset of the data in some cases. We then specified in the text, if we ran the LME on a subset of the data. For example, if the model was run on depths from 50-90 cm, this is specified as $LME_{50-90}$ (e.g. line 422). If the model was run on the whole dataset, it is specified as LME without any subscript (e.g. line 232).*

**Second, please add more justification for why brGDGTs were measured (in terms of specific hypotheses) and how they should be interpreted. The authors mention that they turnover slowly...is this why they were chosen? If so, what does this mean for interpreting the data from a 4-year warming study? Since these are predominantly necromass, is the idea that microbes under heated and control conditions might be preferentially consuming these or warming might accelerate their turnover? Or is the idea to try and see if there is a signal in the brGDGTs that might indicate how microbial communities have overall changed in the past 4 years, which is not visible in the more rapidly cycling PLFAs? Or is the idea to capture predominantly the archaea community, which would not be captured by PLFAs?**

Indeed, we will add some more justification for the use of the brGDGTs, especially in the introduction (line 73) and hypotheses. In the study by Ofiti et al. 2021 (SBB, Warming promotes loss of subsoil carbon through accelerated degradation of plant-derived organic matter) we observed that branched fatty acids were less abundant in warmed plots than control plots. Furthermore, the average chain length of fatty acids was lower in warmed as compared to control plots, especially below 55cm. These observations indicate that there might be decomposition or less input of microbial necromass. The brGDGTs were used as a more time-integrated and independent proxy to assess whether also these compounds, consisting of both microbial biomass, but mostly necromass, are less abundant in warmed plots. This would support the notion that there is either less input or more decomposition of microbial necromass. Thus, the idea is to have an additional proxy for the turnover/input of microbial necromass to soil.

Isoprenoid GDGTs (iGDGTs) are indicative of the archaeal communities. We did not report these values as iGDGTs were low abundant in our samples, for many samples below detection limit.

*We added more justification for the use of brGDGTs in lines 81ff. We further added a hypothesis for the use of brGDGTs in lines 99ff.*

**Minor comments:**
**Since the soil is warmed by a smaller degree in the top 20cm compared to below that, why not just discard the shallow soil data since it cannot fairly be compared with the deeper samples? Also, since lipids were**

**extracted from less soil in shallow compared to deep samples, the deep samples could just be more representative of deep soil and therefore easier to detect a difference in.**

We did observe a similar respiration response at all depths, indicating that the microbial response might be similar despite the difference in temperature magnitude. Nevertheless, we agree that 4°C warming likely affects the microbes and the community differently as compared to only 2.6°C. We do think that reporting all data gives added value, but we propose to discuss this caveat in more depth in paragraph starting on line 318.

We propose to adjust line 319: We did observe a similar respiration response at all depths (Hicks-Pries et al. 2017), indicating that the microbial response might be similar despite the difference in temperature magnitude. Nevertheless, temperature controls the reaction rates of microbial enzymes, which in turn can affect microbial abundance (Allison et al., 2010). Furthermore, incubation experiments show strong effects of temperature magnitude on soil respiration, including subsoil (Yann et al. 2017, Scientific Reports). Thus, the higher magnitude of warming below 20 cm in our experiment might be partially responsible for the observed difference in the microbial response between top- and subsoil.

*We added an extension to the discussion as proposed above. This section can be found in lines 599ff.*

As mentioned by the reviewer, the amount of sample extracted could mean that an effect is easier to be detected in subsoils. We think this is well worth mentioning as a limitation, however would also argue that by thoroughly homogenizing the sample this caveat should be minimized. Furthermore, we also observed more variation in topsoil in parameters where a constant weight was used for the analysis (e.g. carbon), indicating that the variation is likely naturally higher in these topsoils as compared to the subsoil.

**L135: why add the standard in after lipid extraction, and not directly to the soil so that the authors could get a better idea of extraction efficiency for the different soil depths?**

We agree that having a recovery standard can be helpful to have an idea of changing extraction efficiency with depth. However, during implementation of the method in our laboratory we tested having the PC19 standard as a recovery standard alongside the D39C20 standard. We decided to only use the D39C20 standard, because the reproducibility was considerably better when only using the D39C20 standard. Whereas the coefficient of variation was generally below 10% using D39C20, the coefficient was around 20% when using both, D39C20 and the PC19.

**Could figure 4 be presented as an ordination instead? There is a lot to digest here.**

We did use multivariate analysis for the exploration of the data, which did help to identify some of the trends (e.g. actinobacteria, anteiso/iso), but was rather confusing for the purpose of giving a good overview. However, we are happy to reconsider this figure and try and make it more approachable.

*After thorough consideration we did not change the figure to multivariate analysis, but worked on the formatting of the figure. We hope this helps to make the figure easier to digest.*

**Grammar/style**
**The "gram" in Gram positive/negative should be capitalized, as it refers to someone's name (Hans Christian Gram)**

We will adjust this accordingly throughout the manuscript.

*Was changed throughout the manuscript.*

**L177: please mention what kind of post-hoc test was used.**

As mentioned above, we propose to delete the two sections where depth wise Post-hoc tests were conducted and rather focus on the LME results.

*We deleted these sections, as we did not further discuss the post-hoc tests as mentioned above.*

**L281: correlations are generally reported as R (for Pearson) or rho (for Spearman); R^2 is the coefficient of determination.**

We will adjust this accordingly.

*Was changed throughout the manuscript. More specifically in the results sections 3.2, 3.3 and 3.4, as well as in the supplementary figures A2, A3 and A4.*

**L300: there are also a lot of unknowns with respect to extraction efficiency of chloroform fumigation extraction. It almost certainly underestimates the C content of high surface area:volume small cells, as it predominantly captures cytoplasm.**

We will be happy to look into this more and add this limitation with a relevant reference. If the reviewer has a good literature suggestion, this could be very helpful.

**Please also make sure to report the F/T/Z statistic and degrees of freedom, preferably in the text, or otherwise in a supplementary table.**

We will report this information accordingly.

*We added this information throughout the manuscript.*

**Whenever someone says something increased/decreased with depth, it sounds like depth has been treated as a continuous rather than categorical variable. So please try to avoid this.**

Erroneously, we did treat depth as continuous rather than categorical for the analysis. We ran the analysis with depth as categorical and will make the necessary changes. Mostly, the effect of this adaptation was minor, not affecting the interpretation of the results. But in some cases (e.g. overall concentrations of PLFA), the treatment effect did change, leading to minor changes in the manuscript. Find below an overview of the differences for some variables:

*We added the above-mentioned changes to the manuscript.*

**Response to the interactive comment by the anonymous referee # 2**

**The article entitled "Whole soil warming decreases abundance and modifies community structure of microorganisms in subsoil but not in surface soil", address the impact of soil warning in the soil microbial community. The article fits the scope of the journal and it will be of great interest for the journal readers.**

**The introduction summarized the previous knowledge in the topic in clear and concise way with clear and funded hypothesis.**
We thank the reviewer for the constructive and insightful comments. Below we provide our point-by-point reply and we believe the comments will help us further improve the manuscript.

**The material and methods described in detail the laboratory analysis, but the description of the experimental setup is very brief. How was the temperature increase in the soil profile done? The articles you cited would probably describe it in detail and if a reader is interested can go and check those articles; but a short description here (no need to go in deep detail) would help to understand the setup of your experiment.**
We agree that we probably oversimplified the description of the methods section and will add some more detail. As mentioned by the reviewer, a basic understanding of the experimental setup should be possible without necessarily checking additional articles.

We suggest adding the following sentence in line 103: All plots were equipped with vertical steel pipes, which contained resistance heater cables and sand in warmed plots and sand only in control plots. Additionally, two rings of heating cables were buried at 5cm.

*We added the above section, which can be found in lines 135ff.*

**In the result section, the soils depths shown in your tables and figures is not clear. At some point, looks like you sample the 0-5 cm soil surface, and then you sampled in intervals of 10 cm until 85 cm depth, but that is not what I stated in the material and methods. Maybe including the sampling intervals instead of the middle point of it will be clearer. Secondly, when you describe the results in the section 3.1, 3.2, 3.3 and 3.4 the numbers do not match with the respective tables. Are those numbers an average on control and warm soils? If so, this need to be clarify in the text. Regarding the MBC, this data have been used for another paper that has been resubmitted, in my opinion you could discuss about it but not include as a result.**
Generally, we did all analyses in 10cm intervals, so from 0-10cm, 10-20cm,... 80-90cm. It is true we only mention this in the last sentence of the method section and should highlight it better in the figures and tables. We will make these changes accordingly by using sampling intervals as suggested.

*We revised all figures and tables to show the increments as 0-10, 10-20,... We think this does make it less confusing as highlighted by the reviewer.*

We also agree with the second point, as to the mismatch in numbers between text and tables. In the tables we show averages for control and warmed plots, whereas in the text it is the overall averages of control and warmed soils. We will add a clarifying sentence in the text.

*We adapted the numbers in the text. Representative for the trends, we now just mention the values of the control plots.*

The MBC data are only shown in a supplementary figure in the manuscript by Soong et al. (2021) and not evaluated as the difference between control and warmed plots as shown in this manuscript. We think it is a valuable addition to our manuscript to show these data alongside the other microbial proxies and would like to retain the figure as shown.

**Finally, in the discussion section, you mentioned the difference in the heating temperature between the topsoil and the sub-soil in the section 4.2. This should discussed a bit further, since the differences you observed could be easily related to this temperature difference.**
We agree that this issue was not addressed enough and we will add more discussion and raise this concern in the paragraph currently starting with line 318.

We propose to adjust line 319: We did observe a similar respiration response at all depths (Hicks-Pries et al. 2017), indicating that the microbial response might be similar despite the difference in temperature magnitude. Nevertheless, temperature controls the reaction rates of microbial enzymes, which in turn can affect microbial abundance (Allison et al., 2010). Furthermore, incubation experiments show strong effects of temperature magnitude on soil respiration, including subsoil (Yann et al. 2017, Scientific Reports). Thus, the higher magnitude of warming below 20 cm in our experiment might be partially responsible for the observed difference in the microbial response between top- and subsoil.

*We added an extension to the discussion as proposed above. This section can be found in lines 599ff.*

**Below some specific comments**
**Line 21: Gram should start with capital letter**
This will be adjusted accordingly throughout the manuscript.

*We adjusted this throughout the manuscript*

**Line 31: "representative concentration pathway 8.5" this is part of a global model simulation, right? It needs some clarification on the text.**
We agree that the formulation is not concise and will adapt this sentence.
We propose to rephrase line 31 as follows: Global model simulations predict a likely increase of air temperatures of 2.6-4.8°C…

*We rephrased the section accordingly.*

**Line 39: The reference "Fierer et al., 2003b", it should be a Fierer et al., 2003a before in the text. You need to reorganize the references of this author.**
This will be adjusted accordingly.

*We checked the reference guidelines again and it should be correct this way, as the references need to be ordered according to the second authors name.*

**Line 73-77: You introduce the use of the PLFAs with several examples. You should do the same with the brGDGTs.**
We will add some examples in line 75 to have a better justification for the use of brGDGTs. We will further include our hypothesis for the use of brGDGTs as suggested by the other reviewer.

*We added some examples and justification in lines 81ff, as well as a hypothesis in lines 99ff and hope this makes the usefulness of brGDGTs for this study more understandable.*

**Line 83-93: The hypothesis are clear, but I would suggest rewriting a bit this paragraph. Stating at the beginning that you have two main hypothesis and described them, and then add all the text to back them up.**
We will consider this comment in the revisions of the manuscript.

*We reordered the hypothesis and think they should be well understandable now (see lines 94ff). Since we numbered the hypothesis, we also decided to pick them up again in the discussion section (lines 567, 570 and 635, respectively).*

**Line 103: Why was the soil heated at different temperatures along the soil profile? A short explanation should be added.**
The difference in warming magnitude is due to surface heat loss which predominantly affected the top 20cm. We will add this explanation in the text.

*We added a short explanation in line 138.*

**Line 118: Why do you skip some soil depths? I.e. 20-30; 40-50; 60-70; 70-80? A short explanation of why these depths were excluded from the analysis should be added.**
The sample amount was limited, that is why we had to select certain depths only.

*We added a short explanation in line 152.*

**Line 145: This reference is not in the reference list.**
This reference can be found on line 597.

**Line 190: Reference of unpublished articles should not be included.**
This article has now been accepted and published. We will add the full details, including DOI in the list of references.

*We added the missing details.*

**Line 365-367: This paragraph is hanging loose, try to connect to with the previous one.**
We agree this paragraph needs to be connected with the previous one.

*We merged the paragraph with the above section (see lines 657).*

**Line 582-584: Please add the DOI number.**
This will be added.

*We added the missing details.*

**Line 602-603: Please add the DOI number.**
This DOI number was already there, but not well visible because it was moved to the next page.

*We added the missing details.*